# Potential causal association between leisure sedentary behaviors, physical activity and musculoskeletal health: A Mendelian randomization study

Xiaoyan Zhao[1,2], Yan Yang[1,2], Rensong Yue[1]*, Chengguo Su[3]*

**1** Department of Endocrinology, Hospital of Chengdu University of Traditional Chinese Medicine, Chengdu, China, **2** Clinical Medical Department, Chengdu University of Traditional Chinese Medicine, Chengdu, China, **3** Acupuncture and Tuina Department, Chengdu University of Traditional Chinese Medicine, Chengdu, China

* cdzyydxtg@163.com (CS); songrenyue@cdutcm.edu.cn (RY)

**Data Availability Statement:** All relevant data are within the paper and its Supporting Information files.

## Abstract

### Background

Increasing evidence shows that leisure sedentary behaviors (LSB) and physical activity (PA) are associated with various musculoskeletal disorders. However, the causality between LSB/PA and musculoskeletal health remained unknown. In this study, we aimed to evaluate the causal relationships between LSB/PA and lower back pain (LBP), intervertebral disc disorder (IVDD), rheumatoid arthritis (RA), and bone mineral density (BMD) by using a two-sample Mendelian randomization method.

### Methods

The exposure data were obtained from large-scale genome-wide association studies (GWAS), including the PA dataset (self-reported PA, n = 377,234; accelerometer-assessed PA, n = 91,084) and LSB dataset (n = 422,218). The outcome data were derived from the FinnGen LBP dataset (n = 248,528), FinnGen IVDD dataset (n = 256,896), BMD GWAS dataset (n = 56,284), and RA GWAS dataset (n = 58,284). The causal relationships were estimated with inverse variance weighted (IVW), MR-Egger, and weighted median methods. Sensitivity analyses were performed with Cochran's Q test, MR-Egger intercept test, and leave-one-out analysis to estimate the robustness of our findings.

### Results

Genetically predicted leisure television watching increased the risk of LBP (OR = 1.68, 95% CI 1.41 to 2.01; P = 8.23×$10^{-9}$) and IVDD (OR = 1.62, 95% CI 1.37 to 1.91; P = 2.13 × $10^{-8}$). In addition, this study revealed a potential causal relationship between computer use and a reduced risk of IVDD (OR = 0.60, 95% CI 0.42 to 0.86; P = 0.005) and RA (OR = 0.28, 95% CI 0.13 to 0.60; P = 0.001).

**Funding:** This study was supported by the National Natural Science Foundation of China (NO. 82004497) and Xinglin Scholars Project of Chengdu University of Traditional Chinese Medicine (NO. BSH2020009). There was no additional external funding received for this study. The funders had no role in study design, data collection and analysis, decision to publish, or preparation of the manuscript.

**Competing interests:** The authors have declared that no competing interests exist.

## Conclusions

Our results suggest that leisure television watching is a risk factor for LBP and IVDD, whereas leisure computer use may act as a protective factor against IVDD and RA. These findings emphasized the importance of distinguishing between different sedentary behaviors in musculoskeletal disease studies.

## Introduction

Musculoskeletal disorders, including lower back pain (LBP), intervertebral disc disorder (IVDD), rheumatoid arthritis (RA), and osteoporosis, are a major cause of morbidity and disability globally, imposing a heavy economic burden on society [1–4]. IVDD is a common degenerative disorder that is frequently associated with lower back pain [5, 6]. Osteoporosis, a systemic musculoskeletal disease characterized by decreased bone mass and degeneration of bone tissue microarchitecture, has been recognized as one of the most pressing public health problems [7]. Osteoporosis is diagnosed by measuring bone mineral density (BMD) [8]. RA is an autoimmune disease characterized by synovial inflammation as well as bone and cartilage destruction [9, 10]. A key step in the prevention of musculoskeletal disorders is the identification of possible risk factors and protective factors, especially those that can be intervened.

Leisure sedentary behaviors (LSB) refers to any waking behavior characterized by energy expenditure $\leq$1.5 metabolic equivalents in a reclining or sitting posture [11]. Studies have previously shown that sedentary behaviors are associated with increased risk of all-cause mortality, cardiovascular disease, and musculoskeletal disorders [12–15]. Physical activity (PA) is the musculoskeletal movement that results in energy expenditure [16]. There is some evidence that exercise is effective for musculoskeletal pain and confers unique benefits to the musculoskeletal system [17, 18].

Observational studies have suggested that sedentary behaviors may increase the risk of LBP [14, 15] and IVDD [19]. Sedentary behavior and physical inactivity may be associated with poor health-related outcomes (e.g., low functionality and worse disease symptoms) in RA [20]. And studies have indicated that physical activity may reduce the risk of LBP [14], IVDD [21], and osteoporosis [22]. Physical activity interventions for LBP are commonly proposed to work through postural changes and muscle activation [23]. In addition, from a biopsychosocial perspective, LBP involves combinations of different social, psychological, lifestyle, and physical factors [24]. PA is a complex and multidimensional behavior [25]. PA has various positive biopsychosocial effects that are beneficial for the prevention of musculoskeletal system disorders [26].

However, the causal relationship between LSB/PA and musculoskeletal disorder risk is not clear. Existing evidence, based on observational studies, is unable to entirely rule out the possibility of biases, including confounding factors and reverse causality. Besides, randomized controlled studies on this topic are impractical and unethical to constrain participants' physical activity. Mendelian randomization (MR) is a method of using genetic variants associated with the specific exposures of interest to study the causal relationships of modifiable exposures (i.e., possible risk factors) on outcomes [27]. Because single nucleotide polymorphisms (SNPs) are allocated randomly at conception, MR analysis could effectively exclude confounding factors and reverse causality [28]. MR utilizes genetic variations as instrumental variables (IVs) to evaluate the associations between exposures and outcomes.

Given the uncertainty regarding the causal effects of LSB/PA on musculoskeletal disorders, we applied the MR design to assess the potential causal associations. Overall, this MR analysis

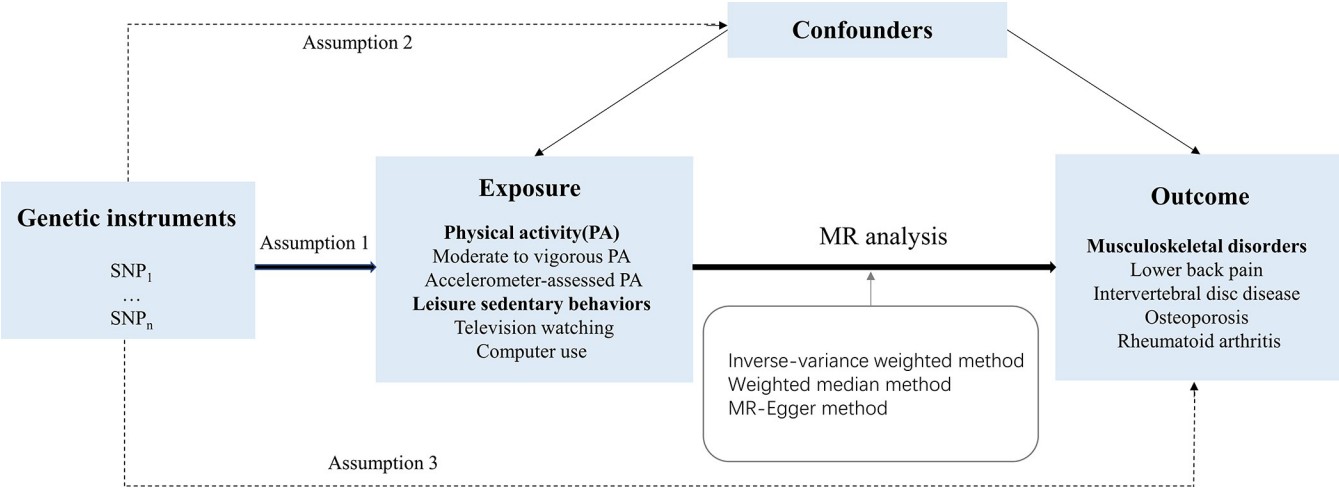

**Fig 1. Study design overview.**

explores the potential causal relationship between LSB/PA and musculoskeletal health (LBP, IVDD, RA, and BMD).

## Methods

### Study design

In this study, we conducted a two-sample MR analysis to investigate the causal associations of PA/LSB with LBP, IVDD, BMD, and RA by using genome-wide association studies (GWAS) summary statistics. To obtain unbiased causal effects, this MR analysis needs to satisfy three assumptions: (i) the genetic variants are robustly related to the exposure of interest; (ii) genetic variants are not associated with potential confounders; and (iii) genetic variants affect the outcomes merely through the exposure of interest. The study design is shown in Fig 1. No separate ethical approval was required in this study.

### Data sources for sedentary behaviors and physical activity

The genetic instruments for sedentary behaviors were obtained from the latest summary-level GWAS of 422,218 European-ancestry individuals from the UK Biobank [29]. The UK Biobank cohort is a large prospective cohort of 500,000 participants recruited in the United Kingdom between 2006 and 2010 [30]. Leisure sedentary behaviors included three phenotypes: hours of television watching per day, hours of computer use per day (excluding computer use at work), and hours of driving per day. The number of driving-related SNPs was insufficient, with a risk of weak instrument bias; therefore, it was not included in our study.

The summary statistics for physical activity were derived from a recent GWAS, which was performed in the UK Biobank [31].This GWAS, consisting of over 377,000 European-ancestry participants, evaluated three measures based on self-report and two measures based on wrist-worn accelerometry data. The self-reported moderate to vigorous physical activity (MVPA) was calculated by taking the sum of total minutes/week of moderate PA multiplied by 4 and the total number of vigorous PA minutes/week multiplied by 8, corresponding to their metabolic equivalents [31]. We used two phenotypes of PA, including self-reported MVPA and accelerometer-assessed PA (average acceleration).

## Data source for LBP, IVDD, BMD, and RA

The summary results of LBP and IVDD were obtained from the FinnGen consortium. On June 1, 2022, FinnGen Data (R7) was released to the public. The dataset contains LBP (21,140 cases and 227,388 controls) and IVDD (29,508 cases and 227,388 controls).

Summarized data of total body BMD was derived from a large meta-analysis of GWAS comprised of 56,284 European-ancestry individuals [8]. Total body BMD (g/cm$^2$) was measured by dual-energy X-ray absorptiometry according to standard manufacturer protocols. Moreover, total body BMD was adjusted for age, weight, height, genomic principal components, and additional study-specific covariates (e.g., recruiting center). The GWAS summary data for total body BMD is obtained from the IEU Open GWAS database [32]. The GWAS ID for BMD is ebi-a-GCST005348.

The summary-level data for RA were available from a large-scale meta-analysis of GWAS, comprising over 100,000 participants of European and Asian ancestries. To reduce population-stratification bias, we extracted summarized data for RA from 58,284 participants of European ancestry (14,361 RA cases and 43,923 controls). A detailed description of the study can be found elsewhere [33]. Summary GWAS data for RA were obtained from the IEU Open GWAS database [32]. The GWAS ID for RA is ieu-a-832. Details of all GWASs included in this study are shown in Table 1.

## Selection of genetic instruments

In this study, we used four sets of genetic instruments indicating LSB and PA, including (1) index SNPs representing television watching (listed in S1 Table in S1 File), (2) index SNPs representing computer use (listed in S2 Table in S1 File), (3) index SNPs representing MVPA (listed in S3 Table in S1 File), and (4) index SNPs representing accelerometer-assessed PA (listed in S4 Table in S1 File).

Genetic instrumental variables were selected by the following criteria: (1) a GWAS-correlated P-value of $5 \times 10^{-8}$; (2) removal of SNPs in linkage disequilibrium (LD, $R^2 \geq 0.001$ and within 10 mb); and (3) removal of SNPs with F statistic less than 10. Instrumental variables with F-statistics <10 were regarded as weak genetic instruments [34].

## Mendelian randomization analyses

Three MR methods, including inverse-variance weighted (IVW), weighted median (WM), and MR-Egger, were conducted to evaluate the causal effects of PA/LSB on LBP, IVDD, BMD, and RA. These MR approaches have different underlying assumptions about horizontal pleiotropy.

**Table 1. Details of the GWASs included in the Mendelian randomization.**

| Consortium | Phenotype | Participants | Ancestry | Web source |
|---|---|---|---|---|
| UK Biobank | Television watching | 422,218 | European | https://pubmed.ncbi.nlm.nih.gov/32317632/ |
| UK Biobank | Computer use | 422,218 | European | https://pubmed.ncbi.nlm.nih.gov/32317632/ |
| UK Biobank | Self-reported MVPA | 377,234 | European | https://gwas.mrcieu.ac.uk |
| UK Biobank | Accelerometer-based PA | 91,084 | European | https://gwas.mrcieu.ac.uk |
| FinnGen | Lower back pain | 248,528 | European | https://www.finngen.fi/en/access_results |
| FinnGen | IVDD | 256,896 | European | https://www.finngen.fi/en/access_results |
| A meta-analysis of GWAS | Total body BMD | 56,284 | European | https://gwas.mrcieu.ac.uk |
| A meta-analysis of GWAS | Rheumatoid arthritis | 58,284 | European | https://gwas.mrcieu.ac.uk |

PA, physical activity; MVPA, moderate to vigorous PA; IVDD, intervertebral disc disorder; BMD, bone mineral density.

IVW, which assumes that instrumental variables can affect the outcome only via the exposure, was used as the main method. Weighted median and MR-Egger methods were implemented as complements to the IVW. The weighted median method provides consistent estimates when more than 50% of the information comes from valid IVs [35]. MR-Egger assumes that the variant-exposure association is independent of the pleiotropic effect of the genetic variant [36].

For important estimates, MR-pleiotropy residual sum and outlier (MR-PRESSO) method was performed to correct horizontal pleiotropy through outlier removal [37]. The MR-Egger intercept, Cochran's Q test, funnel pot, and leave-one-out analyses were used to verify the robustness of our results [38]. Specifically, we assessed horizontal pleiotropy according to the intercept term derived from MR-Egger regression (P < 0.05 was considered the presence of pleiotropy). And P<0.05 of Cochran's Q-test indicated that heterogeneity was detected. We also performed leave-one-out analysis, in which each exposure-related SNP was removed in turn to repeat the IVW analysis, in order to determine whether the causal association was driven by a single SNP.

### Statistics

All analyses were performed using TwoSampleMR (version 0.5.6) in R (version 4.2.1) packages. To account for multiple testing in our analyses, a Bonferroni-corrected P-value was set as 0.05/16 (0.003). P < 0.003 was regarded as statistically significant, and P < 0.05 was considered nominally significant. MR estimates were shown as odds ratios (OR) with 95% confidence intervals (CI) or β estimates with 95% CI. In addition, we calculated the statistical power (https://shiny.cnsgenomics.com/mRnd/).

## Results

### MR estimates

Among the tested LSB phenotypes, IVW analysis indicated that leisure television viewing increased the risk of LBP (OR = 1.68, 95% CI 1.41 to 2.01; P = $8.23 \times 10^{-9}$). Similar causal estimates were gained from WM analysis (OR = 1.54, 95% CI 1.21 to 1.95; P = 0.0004). The result from MR-Egger analysis showed a consistent but nonsignificant direction (OR = 2.31, 95% CI 1.00 to 5.34; P = 0.055) (Fig 2A). Additionally, a significant association between computer use and LBP was observed using IVW analysis, whereas the causal estimates from MR-Egger analysis displayed directional inconsistency (Fig 2A).

Television viewing was significantly associated with an increased risk of IVDD with the IVW method (OR = 1.62, 95% CI 1.37 to 1.91; P = $2.13 \times 10^{-8}$). Meanwhile, similar risk estimates were obtained using the WM method (OR = 1.56, 95% CI = 1.27 to 1.93, P = $3.19 \times 10^{-5}$). MR-Egger also showed similar causal estimates, although the association was not statistically significant (Fig 2B). In addition, a nominally significant association between computer use and IVDD was observed using IVW analysis (OR = 0.60, 95% CI 0.42 to 0.86; P = 0.005). WM and MR-Egger showed a consistent but nonsignificant association (Fig 2B).

Among the LSB phenotypes (television watching and computer use), we did not observe evidence of causal associations between genetically predicted sedentary behaviors and total body BMD (Fig 2C).

Moreover, a negative causality between genetically proxied computer use and RA was detected using IVW analysis (OR = 0.28, 95% CI 0.13 to 0.60; P = 0.001). WM analysis showed a nominally significant result (OR = 0.26, 95% CI 0.10 to 0.67; P = 0.006) and MR-Egger presented a consistent but nonsignificant result (Fig 2D). Additionally, causal

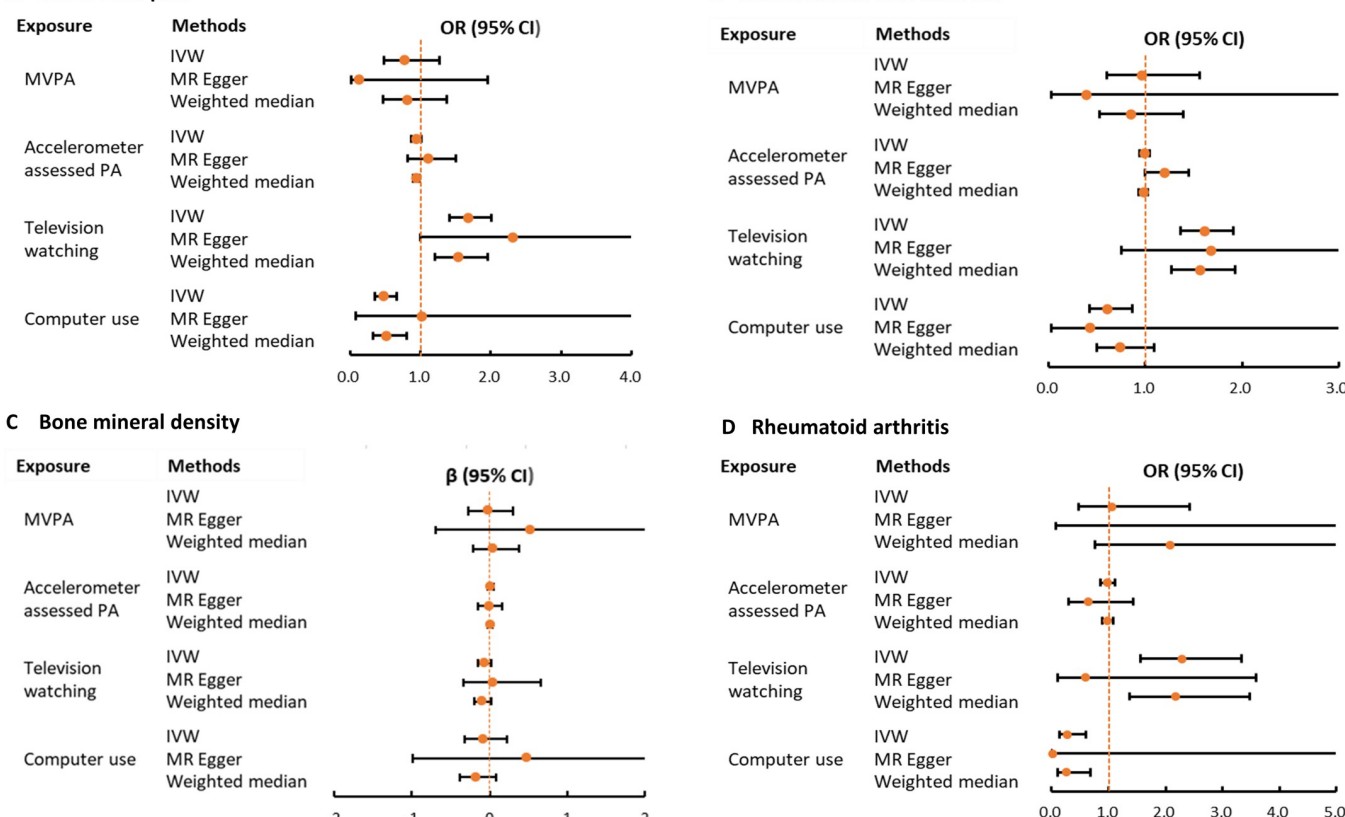

**Fig 2.** Causal effects for PA/LSB on lower back pain (A), intervertebral disc disorder (B), bone mineral density (C) and rheumatoid arthritis (D). Summarized Mendelian randomization (MR) effect sizes were derived from the inverse variance weighted (IVW), MR-Egger and weighted median methods. PA, physical activity; MVPA, moderate to vigorous physical activity.

estimates of the relationship between television watching and RA displayed directional inconsistency in three MR methods (Fig 2D).

Furthermore, among the tested PA phenotypes (MVPA and accelerometer assessed PA), we did not observe evidence of causal associations between genetically proxied PA phenotypes and musculoskeletal disorders (LBP, IVDD, BMD, and RA) (Fig 2A–2D). In addition, details on MR estimates of the causal relationships between PA/LSB and these musculoskeletal disorders are shown in S5 Table in S1 File. We also calculated the statistical power for important estimates. A power of over 80% was observed. Statistical power is presented in S6 Table in S1 File.

## Sensitivity analyses

The Cochran's Q test and the MR-Egger intercept test were conducted to evaluate the robustness of the results (Table 2). The P-values of the MR-Egger intercept tests were all > 0.05, suggesting that there was no horizontal pleiotropy. However, heterogeneity was detected in the Cochran's Q test except for estimates of MVPA on RA (P = 0.093), accelerometer assessed PA on RA (P = 0.068), television watching on BMD (P = 0.128), computer use on LBP (P = 0.330), and computer use on RA (P = 0.081). Although heterogeneity was observed in some results, it did not invalidate the MR estimates as random-effect IVW in the current study. Furthermore, the Egger intercept did not detect any pleiotropy, suggesting that no pleiotropy bias was

**Table 2. Sensitivity analysis of the causal effects of LSB/PA on musculoskeletal disorders.**

| Exposure | Outcome | Cochran Q test | | MR-Egger | |
|---|---|---|---|---|---|
| | | Q value | P | Intercept | P |
| MVPA | Lower back pain | 32.117 | 0.015 | 0.027 | 0.204 |
| | Intervertebral disc disorder | 40.357 | 0.001 | 0.013 | 0.525 |
| | Bone mineral density | 38.448 | 0.003 | -0.008 | 0.403 |
| | Rheumatoid arthritis | 23.854 | 0.093 | -0.030 | 0.394 |
| Accelerometer assessed PA | Lower back pain | 25.390 | 0.001 | -0.043 | 0.329 |
| | Intervertebral disc disorder | 17.617 | 0.014 | -0.051 | 0.080 |
| | Bone mineral density | 19.198 | 0.008 | 0.004 | 0.848 |
| | Rheumatoid arthritis | 8.737 | 0.068 | 0.116 | 0.373 |
| Television watching | Lower back pain | 105.958 | 0.019 | -0.005 | 0.453 |
| | Intervertebral disc disorder | 124.447 | 0.001 | -0.001 | 0.931 |
| | Bone mineral density | 103.236 | 0.128 | -0.002 | 0.600 |
| | Rheumatoid arthritis | 100.156 | 0.010 | 0.018 | 0.199 |
| Computer use | Lower back pain | 21.123 | 0.330 | -0.012 | 0.567 |
| | Intervertebral disc disorder | 30.211 | 0.035 | 0.006 | 0.807 |
| | Bone mineral density | 57.519 | 0.00002 | -0.009 | 0.634 |
| | Rheumatoid arthritis | 24.409 | 0.081 | 0.053 | 0.322 |

LSB, leisure sedentary behaviors; PA, physical activity; MVPA, self-reported moderate to vigorous physical activity.

introduced into the MR estimates despite heterogeneity (Fig 3). Funnel plots, forest plots, and leave-one-out analyses are shown in S1-S8 Figs in S1 File. The results were not biased by a single SNP, indicating that estimates were not violated.

## Discussion

In the current study, we evaluated the casual associations between genetically predicted LSB/PA phenotypes and different musculoskeletal disorders (LBP, IVDD, BMD, and RA) by using three MR methods. Our results suggest that leisure television watching is a risk factor for LBP and IVDD, whereas leisure computer use may be a protective factor against IVDD and RA. In addition, we found little evidence for the association between PA phenotypes and musculoskeletal disorders.

Observational studies have suggested that sustained sedentary behaviors are negatively associated with musculoskeletal health [15, 39]. Lack of exercise is a main cause of most chronic diseases, including chronic musculoskeletal disorders [40]. However, direct evidence for the causal relationships between LSB/PA and various musculoskeletal disorders is still lacking. Compared with large-scale prospective clinical trials requiring long-term observation, the MR study revealed the potential causal relationship between LSB/PA and different musculoskeletal disorders in a time- and cost-saving manner.

Television watching is the main sedentary behavior associated with leisure time [41]. Compared with other leisure sedentary traits, television watching was a typical sedentary behavior, leading to an energy surplus through lower energy expenditure and excessive energy intake (especially snacks) [42]. Studies have shown that weight gain associated with an energy surplus is a risk factor for LBP and IVDD [43, 44]. Consistent with these findings, our MR results indicated a potential causal association of television watching with LBP and IVDD. Intriguingly, leisure computer use was negatively associated with IVDD and RA. These results indicated a biological heterogeneity behind different sedentary behaviors. In addition, MR-Egger intercept

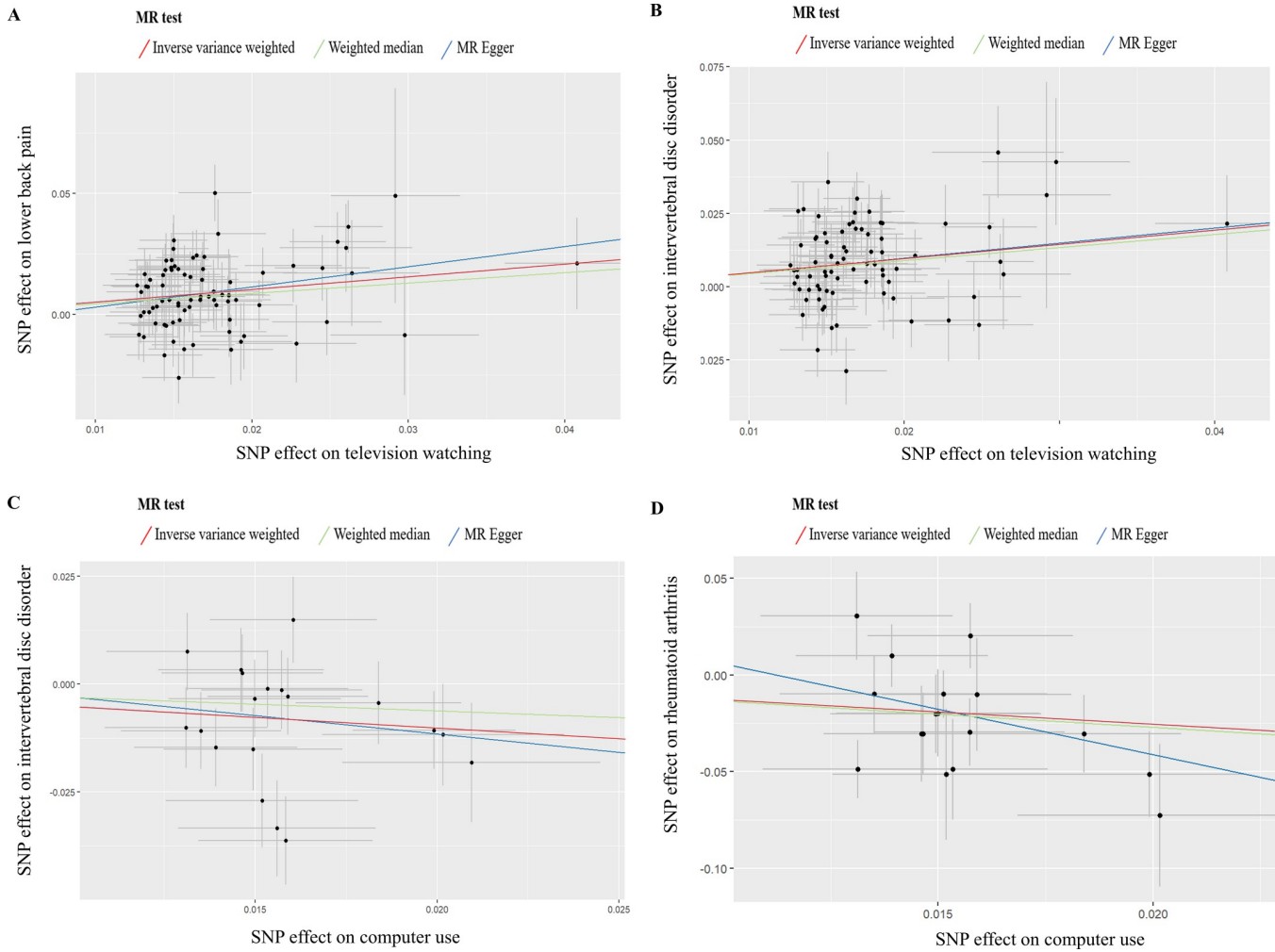

**Fig 3.** Scatter plots from genetically predicted television watching on lower back pain (A), television watching on intervertebral disc disorder (B), computer use on intervertebral disc disorder (C) and computer use on rheumatoid arthritis (D).

tests detected no horizontal pleiotropy. Sensitivity analyses suggested the robustness of our results. Moreover, the causal inference had high reliability because the statistical power of the IVW method for important estimates was above the threshold of 80%.

Additionally, our in-depth analyses provided several potential explanations for the causal relationships between sedentary behaviors and LBP, IVDD, and RA. First, television watching is generally considered 'mentally passive' behavior, whereas computer use is usually considered 'mentally active' behavior [45, 46]. Mentally passive sedentary behavior (television watching) was always accompanied by poor physical and mental health (such as depression) [45, 46]. A meta-analysis showed that symptoms of depression increased the risk of developing LBP [47]. And depression is strongly associated with RA [48]. A recent study illustrated the bidirectional communication between the central nervous system and chronic peripheral inflammation, which indicated the pathophysiologic interactions between neuropsychiatric diseases and RA, highlighting the existence of the joint-brain axis [9]. Therefore, mental health disorders might serve as a key intermediate factor in the LSB-RA pathway. Second, sitting behavior is associated with LBP [49]. Patients with LBP exhibited a more static sitting behavior. Compared with static posture, dynamic sitting behavior is thought to reduce spinal load

and muscle fatigue [50, 51]. Television viewing is a more immersive and less reflective form of leisure entertainment than computer use [52]. Television watchers tend to show a more static sitting behavior, which is a risk factor of LBP and IVDD. Third, older adults are more likely to watch television as a leisure entertainment, whereas young people tend to spend more time on computer use [52]. Aging has been considered an important cause of musculoskeletal disorders, especially LBP and IVDD [5, 53, 54]. Our study indicates that public health initiatives targeting LSB can lower the risk of LBP and IVDD, thereby gaining potential health benefits. The major audience for television watching, especially elderly adults, should be assisted in reducing television viewing time and static sitting behavior.

This MR study has several strengths. The main strength of our study is the two-sample MR design, which reduces unobserved confounding and reverse causality. Additionally, multiple sensitivity analyses were performed to test the validity of the MR assumptions, minimizing the possibility of biased results. IVW was used as the main method, with higher statistical power than other MR methods, especially MR-Egger [52, 55]. Thus, it is not surprising that the MR-Egger results with low statistical power had nonsignificant P values and wider confidence compared to IVW. Our study strengthened the requirement for consistent beta direction in all MR methods.

This MR study has several limitations. First, the findings cannot be immediately applied to other ethnic groups with different cultures and lifestyles, as the participants in the datasets were of European ancestry. Second, it is difficult to completely exclude pleiotropy because the functional biological roles of these genetic variants are not fully understood. Horizontal pleiotropy affects the stability of MR results, but vertical pleiotropy, where exposure acts on outcome through other factors along the same causal pathway, is acceptable. Third, our findings show the potential causal relationships between different sedentary behaviors and musculoskeletal disorders, but the potential mechanisms warrant further study.

## Conclusions

Our results suggest that television viewing is causally related to an increased risk of LBP and IVDD, while computer use may be causally associated with a decreased risk of IVDD and RA. Our findings highlight the importance of distinguishing between different domains of sedentary behaviors.

## Supporting information

**S1 File.**
(DOC)

## Acknowledgments

We wish to acknowledge all participants and investigators who contributed to the GWAS data.

## Author Contributions

**Writing – original draft:** Xiaoyan Zhao.

**Writing – review & editing:** Yan Yang, Rensong Yue, Chengguo Su.

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
