## [Decision Letter · Decision Letter 0]

23 Jan 2023

PONE-D-22-33436Causal association between leisure sedentary behaviors, physical activity and musculoskeletal health: a Mendelian randomization studyPLOS ONE

Dear Dr. Su,

Thank you for submitting your manuscript to PLOS ONE. After careful consideration, we feel that it has merit but does not fully meet PLOS ONE’s publication criteria as it currently stands. Therefore, we invite you to submit a revised version of the manuscript that addresses the points raised during the review process.

We look forward to receiving your revised manuscript.

Kind regards,

Renato Polimanti, Ph.D.

Academic Editor

PLOS ONE

Journal Requirements:

"This project was supported by the National Natural Science Foundation of China (8200153156) and Xinglin Scholars Project of Chengdu University of Traditional Chinese Medicine (BSH2020009)."

"This project was supported by the National Natural Science Foundation of China (8200153156) and Xinglin Scholars Project of Chengdu University of Traditional Chinese Medicine (BSH2020009).The funders had no role in study design, data collection and analysis, decision to publish, or preparation of the manuscript."

We note that one or more of the authors is affiliated with the funding organization, indicating the funder may have had some role in the design, data collection, analysis or preparation of your manuscript for publication; in other words, the funder played an indirect role through the participation of the co-authors. If the funding organization did not play a role in the study design, data collection and analysis, decision to publish, or preparation of the manuscript and only provided financial support in the form of authors' salaries and/or research materials, please do the following:

a. Review your statements relating to the author contributions, and ensure you have specifically and accurately indicated the role(s) that these authors had in your study. These amendments should be made in the online form.

b. Confirm in your cover letter that you agree with the following statement, and we will change the online submission form on your behalf: 

“The funder provided support in the form of salaries for authors [insert relevant initials], but did not have any additional role in the study design, data collection and analysis, decision to publish, or preparation of the manuscript. The specific roles of these authors are articulated in the ‘author contributions’ section.”"

Additional Editor Comments:

although reviewers appreciated the manuscript, they also found several major issues that should be addressed.

To be considered for publication, the authors should fully address the problems highlighted by reviewers in a revised submission.

Reviewers' comments:

Reviewer's Responses to Questions

**Comments to the Author**

1. Is the manuscript technically sound, and do the data support the conclusions?

Reviewer #1: Yes

Reviewer #2: Partly

2. Has the statistical analysis been performed appropriately and rigorously? 

Reviewer #1: Yes

Reviewer #2: Yes

3. Have the authors made all data underlying the findings in their manuscript fully available?

Reviewer #1: Yes

Reviewer #2: Yes

4. Is the manuscript presented in an intelligible fashion and written in standard English?

Reviewer #1: Yes

Reviewer #2: Yes

5. Review Comments to the Author

Reviewer #1: Summary Paragraph

The authors present a Mendelian Randomization study assessing the potential causal associations between leisure sedentary behaviors and physical activity phenotypes and a variety of musculoskeletal disorders such as lower-back pain, intervertebral disc disorder, rheumatoid arthritis, and bone mineral density. This work has the potential of increasing the understanding of the potential causal relationships between known risk factors for musculoskeletal disorders and this outcome using genetically based instrumental variables. One of my main concerns is the use of strong causal language in the title, discussion section and conclusions. Another concern is the reduced scope of interpretations of these results in the discussion (more extensive explanations about biological mechanism [e.g., from SNPs to tissues], and about potential clinical relevance would be needed).

Strengths:

The authors explained well the background of the study and justification for using MR. Exposure and outcome variables are clear in the introduction and throughout the manuscript. The authors outlined well how the exposure and outcome variables have a plausible relationship. The assumptions and the study design were described clearly and tested, quality control and selection of genetic instruments were also well outlined.

Weaknesses or areas for improvement:

Since biological mechanisms, their interaction across time, type and time of measurements and MR assumptions play a crucial role here, the use of strong causal language in the title, discussion section and conclusions could undermine the words of caution when interpreting results from MR analyses.

Points for clarification are outlined below by section.

Title:

• I’d suggest the authors to add the word “Potential” before “Causal association (…)” in the title.

Abstract:

• (p. 8) Usually MR hypotheses include causal language or language inferring the potential causality. In the abstract’s background, the authors use “related”, which hints more to non-directional associations in their hypothesis.

Introduction:

• (p. 9) I'd suggest to clarify "It" after the reference call #7. Although it can be assumed what the authors are talking about, it is usually best to specify the subject of the new sentence, especially after a long multi-clause one.

• (p. 9) Add a few examples of the unique benefits of exercise for the musculoskeletal system.

Method:

• (p. 10) Although it might be assumed that the participants for the two phenotypes of PA have European ancestry, specify if this is the case in the corresponding paragraph under the 2.2 subsection.

• (p. 11) Add an “Ancestry” column in Table 1 if multiple ancestries were used. A column listing the data sources for each Consortium could also be helpful for Table 1 (or this list could also be in the notes part of Table 1 to save space).

• (p. 12) It would be interesting to inspect the differences of significance from unadjusted p-values to at least two multiple-testing adjustments. The authors included the Bonferroni adjustment, which is conservative and may increase Type-II error. Would the authors be able to estimate p-values under a less conservative method (e.g., false discovery rate adjustment) and add them next to the Bonferroni adjusted ones?

• Were any power calculations performed prior to the analysis? Whether these calculations were performed or not, it would be interesting to see the differences in statistical power from each sample.

• Are the authors able to provide more information about the covariates beyond BDM? And mention whether these covariates were adjusted for across samples or not?

• Would the authors be able to provide information about the characteristics of missing data? And if present, how the missingness was addressed?

• Were this study and its protocols pre-registered? If so, where can the reader find them?

Results:

• Since this study used a two-sample MR approach, are the authors able to include information about the variant-exposure correlations between the samples of exposure and outcome phenotypes? And, would the authors also be able to include any report of participant overlap between exposure and outcome samples?

• Did the authors test the direction of causation in the opposite direction (i.e., outcome -> exposure, or bidirectional MR)? Many biological mechanisms are complex and might include feedback loops that may hint scientist to test the plausibility of the reverse direction of causation (e.g., causal influences of outcome on exposure at unmeasured timepoints). If the authors tested this direction, would they be able to provide the results? If they didn’t test it, would they be able add this supplementary analysis to test this direction?

Discussion:

• Would the authors be able to discuss any potential clinical relevance and public health applications of these results beyond the discussion provided?

• The authors mentioned in the limitations paragraph that these results cannot be generalized to populations of other ancestries different than European, which is true. Are the authors able to discuss the importance of expanding the genetic and phenotypic research more to non-European ancestries? If so, would they inform the reader about their effort to find datasets/summary-stats of non-European ancestry for their analyses in this manuscript?

• (p. 15-16) I’d suggest to add “potential” before “causal relationship” in the following sentence “(…), the MR study revealed the causal relationship between LSB/PA and different musculoskeletal disorders (…)” and also in the many occurrences of “causal relationship” or “causal association” or other similar assertions throughout the discussion section and also importantly, in the Conclusions. It is always important to use causal language cautiously in MR studies.

Figures:

• This might be an issue of the pdf convertor, but I’d encourage the authors to double check that figures (and any image-based object/file) do not pixelate. Otherwise reading small letters or visualizing shapes can become difficult. I suggest to use vector-based images.

Other:

• Is there a link where readers can find the code used for the analyses? This can enhance transparency and reproducibility.

Reviewer #2: 1. The GWAS’s used have all been on individuals with European ancestry. I would request the authors to replicate these findings or at least perform these analyses in GWAS’s of other ancestries.

2. I would like the authors to perform the bi-directional MR also i.e. from musculoskeletal disorders to LSB/PA

3. It is not clear to me why did the authors select RA, LBP, IVDD for musculoskeletal disorders? Were they just examples? Why did the authors not select carpal tunnel syndrome and tendonitis?

4. I would like the authors to elaborate on the sensitivity analyses and the results in the discussion section a bit more.

6. PLOS authors have the option to publish the peer review history of their article (what does this mean?). If published, this will include your full peer review and any attached files.

Reviewer #1: No

Reviewer #2: No

---

## [Author Response · Author response to Decision Letter 0]

6 Feb 2023

We appreciate the editors and reviewers very much for their comments and suggestions on our manuscript entitled“Potential causal association between leisure sedentary behaviors, physical activity and musculoskeletal health: a Mendelian randomization study”. We have made revisions which are marked in red in the article. The revised manuscript has been uploaded.

Journal Requirements:

1. We revise the article according to PLOS ONE's style.

2. We ensure that we provide the correct ‘Funding Information’.

3. We have added the statement “There was no additional external funding received for this study.”

4. "This project was supported by the National Natural Science Foundation of China (8200153156) and Xinglin Scholars Project of Chengdu University of Traditional Chinese Medicine (BSH2020009). The funders had no role in study design, data collection and analysis, decision to publish, or preparation of the manuscript."

5. We added the ORCID iD for the corresponding author.

Reviewer #1

1. We have revised the article according to the suggestion. We have added a few examples of the unique benefits of exercise. We have added an “Ancestry” column in Table 1. We calculated the statistic power for important estimates. A power of over 90% was observed. Statistical power was presented in Table S6.

2. GWAS data about exposures are all from the UK Biobank，but GWAS data about outcomes were obtained from the FinnGen consortium and meta-analysis of GWAS. Therefore, there is very little sample overlap and it did not invalidate the MR results.

3.Considering that LBS and PA belong to behaviors instead of diseases, it may not be appropriate to use them as outcome phenotypes. Therefore, we did not perform a bidirectional MR.

4. To reduce population-stratification bias, we extracted summarized data of European ancestry. This is a strength of our study. Meanwhile, it limits the generalizability of our findings to other ethnic groups. At present, no aggregated data from genome-wide association studies of non-European ancestry are available for this MR study.

5. We checked these figures and resized the figures. In addition, data source and the link are shown in the article.

Reviewer #2

1.To reduce population-stratification bias, we extracted summarized data of European ancestry. This is a strength of our study. Meanwhile, it limits the generalizability of our findings to other ethnic groups. At present, no aggregated data from genome-wide association studies of other ancestries are available to conduct the MR study. Further studies will be done if data are available for other ancestries.

2. Considering that LBS and PA belong to behaviors instead of diseases, it may not be appropriate to use them as outcome phenotypes. Therefore, we did not perform a bidirectional MR.

3. First, we selected several common musculoskeletal disorders. This study shows the association between LBS/PA and RA, LBP, IVDD, and BMD, but it cannot describe the association of LBS/PA with other musculoskeletal disorders. Second, observational studies have suggested that LBS/PA are associated with RA, LBP, IVDD, and BMD, so we selected these diseases.

4. We added more description in the discussion section.

---

## [Decision Letter · Decision Letter 1]

1 Mar 2023

Potential causal association between leisure sedentary behaviors, physical activity and musculoskeletal health: a Mendelian randomization study

PONE-D-22-33436R1

Dear Dr. Su,

We’re pleased to inform you that your manuscript has been judged scientifically suitable for publication and will be formally accepted for publication once it meets all outstanding technical requirements.

Kind regards,

Renato Polimanti, Ph.D.

Academic Editor

PLOS ONE

Additional Editor Comments (optional):

The authors adequately addressed reviewers' concerns and the manuscript is now suitable for publication in PLOS One.

Reviewers' comments:

Reviewer's Responses to Questions

**Comments to the Author**

1. If the authors have adequately addressed your comments raised in a previous round of review and you feel that this manuscript is now acceptable for publication, you may indicate that here to bypass the “Comments to the Author” section, enter your conflict of interest statement in the “Confidential to Editor” section, and submit your "Accept" recommendation.

Reviewer #1: All comments have been addressed

2. Is the manuscript technically sound, and do the data support the conclusions?

Reviewer #1: Yes

3. Has the statistical analysis been performed appropriately and rigorously? 

Reviewer #1: Yes

4. Have the authors made all data underlying the findings in their manuscript fully available?

Reviewer #1: Yes

5. Is the manuscript presented in an intelligible fashion and written in standard English?

Reviewer #1: Yes

6. Review Comments to the Author

Reviewer #1: (No Response)

7. PLOS authors have the option to publish the peer review history of their article (what does this mean?). If published, this will include your full peer review and any attached files.

Reviewer #1: No

---

## [Editor Report · Acceptance letter]

7 Mar 2023

PONE-D-22-33436R1 

Potential causal association between leisure sedentary behaviors, physical activity and musculoskeletal health: a Mendelian randomization study 

Dear Dr. Su:

I'm pleased to inform you that your manuscript has been deemed suitable for publication in PLOS ONE. Congratulations! Your manuscript is now with our production department. 

Kind regards, 

on behalf of

Dr. Renato Polimanti 

Academic Editor

PLOS ONE